# The Male Experience of Suicide Attempts and Recovery: An Interpretative Phenomenological Analysis

**DOI:** 10.3390/ijerph18105209

**Published:** 2021-05-14

**Authors:** Cara Richardson, Adele Dickson, Kathryn A. Robb, Rory C. O’Connor

**Affiliations:** 1Institute of Health & Wellbeing, University of Glasgow, Glasgow G12 0XH, UK; Katie.Robb@glasgow.ac.uk (K.A.R.); Rory.OConnor@glasgow.ac.uk (R.C.O.); 2Centre for Applied Behavioural Sciences (CABS), Heriot-Watt University, Edinburgh EH14 4AS, UK; A.Dickson@hw.ac.uk

**Keywords:** suicide, men, males, risk, attempt, recovery, protective factors

## Abstract

Suicidal behaviour is a complex phenomenon—its aetiology spans biological, psychological, environmental, social and cultural facets. Men’s deaths by suicide outnumber women in every country in the world. This study explored the male experience of suicide attempts and recovery as well as factors which may be protective for men. Men (n = 12) participated in semi-structured face-to-face interviews which were subjected to Interpretative Phenomenological Analysis (IPA). Four master themes were identified: (1) “characteristics of attempt/volitional factors”, (2) “dealing with suicidal thoughts and negative emotions”, (3) “aftermath” and (4) “protective factors”. The theoretical and clinical implications of this study are discussed, including help seeking, emotional expression, the long-term impact of suicide attempt as well as the applied contribution to established theories.

## 1. Introduction

Suicidal behaviour is a complex phenomenon—its aetiology spans biological, psychological, environmental, social and cultural facets [1,2]. Male deaths by suicide outnumber those by women in every country in the world [3]. The gender paradox of suicide describes the fact that women are more likely to attempt suicide, but men are more likely to die by suicide [4]. There have been several potential explanations proposed to account for this, including that men may experience or display signs of mental illness differently to women, such as displaying more aggressive or avoidant behaviours [5], which then may predispose them to self-injurious or risky behaviours. How men cope with difficult life events such as relationship breakdown or unemployment is also relevant [6], similarly linked to engaging in risky behaviours. Moreover, differences in method of suicidal behaviour among men and women may have an influence—for example, men are more likely to use more lethal methods such as firearms [7]. A “failed” suicide may be viewed as weak and a threat to masculinity, whereas a “successful” suicide is viewed as brave and decisive [4]. However, despite the difference in methods, men and women may not differ in terms of suicidal intent [8]. 

Men may also have difficulties recognising that they are in distress and misinterpret changes in their thoughts and behaviour. Player et al [9] explained that men may not make the connection between their mood, behaviours and suicide risk. Previous research has also identified particular barriers towards seeking help, among young men, including a fear of being diagnosed with a mental illness, feeling there is “no room for weakness” and intense shame [10]. This may manifest itself as masking emotions and withdrawal from relationships before their death either to protect themselves from being rejected or protect their partner/family member from the pain of losing them to suicide [10]. Danielsson and Johansson [11] also found that men often felt more comfortable describing symptoms of mental illness in terms of physical symptoms as opposed to emotional symptoms. 

Methods of support particularly relevant to men who had survived a suicide attempt include distraction as well as practical, emotional and professional support [9]. Providing men with practical support, particularly managing a crisis, may halt the progression from suicidal ideation to behaviour [9]. An enhanced understanding of the male experience of suicidal thoughts and behaviours can also aid support networks in responding to the needs of their loved ones [12]. Fear of being a burden to friends or family and being isolated from others have also been identified as barriers to seeking help in men [13]. By having support from other people that they trust, respect and feel they can relate to, men may feel listened to and be more likely to access help and support on their terms [13]. Reminders of the impact that their death would have on their family may also be significant [9,13]. 

Previous research has investigated the antecedents to suicidal behaviour in men such as work pressures [14,15], access to means [14] and difficulties in the family or romantic relationships [15,16,17]. However, there is a smaller literature base on how men cope and recover following a suicide attempt [18]. The present study aims to address this gap by exploring the male experience of suicide attempts and recovery. By interviewing men with a history of suicidal thoughts and behaviours, this study allows for the exploration of the antecedents of their suicide attempt and the impact thereafter. The factors that may be protective for men in suicidal crisis are also explored. 

## 2. Materials and Methods

### 2.1. Sampling

A sample of twelve men who had attempted suicide in the past five years was recruited through social media adverts (Twitter, Facebook, Gumtree and University website). Attempted suicide was defined as having engaged in a non-fatal, self-directed self-harming episode associated with at least some evidence of suicide intent [19]. Inclusion criteria were identifying as male; at least 18 years old; having attempted suicide in the last five years; and being competent in English. Exclusion criteria were being imminently suicidal (i.e., a person stating that they intended to kill themselves within the next few hours); experiencing a psychotic episode at the time of recruitment; and having a suicide attempt more than five years ago. A total of 31 men were screened for eligibility over the telephone and 12 met the eligibility criteria for participation. Participants were aged between 19 and 49 years (M = 33.8, SD = 9.8), and were from Scotland (UK). Recruitment took place between April and July 2019. Among the 12 participants, five men had attempted suicide in the last year. The age at which participants first thought about suicide ranged from 12 to 44 years (M = 19.9, SD = 9.5) and the age when they first attempted suicide varied between 12 and 44 years (M = 23.8, SD = 8.8). Further details on participants’ demographic information and suicidal history can be found in Appendix A Table A1.

### 2.2. Procedure and Interview 

Ethical approval for this study was obtained from the ethics committee of the College of Medical, Veterinary and Life Sciences (MVLS) at the University of Glasgow (reference: 200180116). Potential participants contacted the author via text, telephone call, email or social media and scheduled an eligibility screening phone call. Before the telephone call, the information sheet and a support sheet, with a list of organisations to contact if participants wished to seek support (for example, Samaritans, Breathing Space and Scottish Association for Mental Health), were emailed to all potential participants. The potential participants were also given the opportunity to ask any questions about this study during the phone call.

Following telephone screening, and if the eligible participants were still interested in participating in this study, a face-to-face interview was arranged at their convenience. Semi-structured interviews were conducted by the first author either at the Suicidal Behaviour Research Lab or the Scottish Association for Mental Health (SAMH) offices. The interviews were audio-recorded and lasted between 25 and 67 min (M = 44.1 min). No one else was present besides the interviewer and participant. All participants were offered £30 compensation for their time. A brief interview schedule was created based on the overall aim of this study. This began with “Tell me about your most recent experience of attempting to take your own life”. Relevant topics were then explored with follow-up questions such as “How did that make you feel?” and “What was going through your mind at that time?”. This semi-structured process helped the interviewer guide the participant through the process, without asking leading questions. The interviewer also used some reflection and probing techniques (such as “You mentioned… can you tell me a bit more about that?”). A risk assessment was conducted after the interviews to ensure participants’ safety—this included clinical measures of psychological distress and suicidal intent. No participants indicated feelings of distress following the interviews. The transcripts were not shared with the participants prior to or following the analysis. The participants all opted into being sent the results from this study following publication. The focus of this paper is on attempts and recovery; other themes, such as “social expectations of being a man”, were also identified from this data and are included in another paper (in preparation).

### 2.3. Analysis

The interviews were analysed using Interpretative Phenomenological Analysis (IPA) [20,21]. IPA is a detailed examination of the human lived experience and is concerned with each participant’s lived experience of a specific event (phenomenology), their attitudes towards the event, and the significance placed on this and their account of this experience (idiographic account) [21]. Due to the in-depth nature of IPA, a small sample size is advised. The process of conducting an IPA involves hermeneutics, it is a deeply interpretative process, and the preconceptions of the researcher are considered during analysis.

The steps detailed by Smith and Shinebourne [21] were undertaken and included (1) the close, line by line analysis of the experiential claims, concerns and understandings of each participant; (2) the identification of the emergent patterns (i.e. themes) within this experiential material, emphasising both convergence and divergence, commonality and nuance, usually first for single cases, and then subsequently across multiple cases. Then, (3) the development of a “dialogue” between the researchers, their coded data, and their psychological knowledge about what it might mean for participants to have these concerns, in this context, leading in turn to the development of a more interpretative account; (4) the development of a structure, frame or gestalt which illustrates the relationships between themes. Following this, (5) the organisation of all of this material in a format which allows for analysed data to be traced through the process, from initial comments on the transcript, through initial clustering and thematic development, into the final structure of themes; (6) the use of supervision, collaboration, or audit to help test and develop the coherence and plausibility of the interpretation. Finally, (7) the development of a full narrative, evidenced by a detailed commentary on data extracts, which takes the reader through this interpretation, usually theme by theme, and is often supported by some form of visual guide (a simple structure, diagram or table) and (8) reflection on one’s perceptions, conceptions and processes.

### 2.4. Research Team and Reflexivity 

The interviews were conducted by the first author, who is a female PhD student who has a first degree in psychology. This study was supervised by the co-authors: the second author is an IPA expert; the third author is a health psychology researcher; and the fourth author is a health psychologist who has been researching suicide for more than 20 years. A sample of the transcripts was sent to the supervisors for independent analysis as well as discussion and agreement on themes. The first author also sought credibility checking from a supervisor regarding interview coding. There was no relationship established between the researcher and participants before the commencement of this study. The only information disclosed to the participants about the research was the institutional affiliation and that she was conducting a study on risk factors for suicidal behaviour in men. There were no characteristics of the interviewer reported. 

## 3. Results

### 3.1. Overview 

Four master themes were identified related to the male experience of suicide attempts and recovery. These will be explored alongside the related sub-themes (Table 1) and each theme will be supported by a verbatim quote from the interview transcripts. Minor edits were made to the quotes, translating regional dialect whilst retaining the original terms used in brackets. Additional quotes are also analysed in Appendix B.

### 3.2. Characteristics of Attempt/Volitional Factors

#### 3.2.1. Change in Thinking 

All of the men (n = 12) interviewed described how in the lead up to the attempt, they experienced a shift in their pattern of thinking, that once they had decided that they were going to take their own life, there was nothing that was going to stop them:


*"and that was the first thing that I thought because I thought right that’s going to be the sharpest thing that’ll do the job… umm, I was thinking… I wasn’t thinking rationally in that side but in the in the mechanics of doing it… I was sort of thinking ehh very clear…and methodical in that way…and… the feelings and things they were just all over the place…I wasn’t thinking clearly"*


In the face of chaotic and difficult emotions and feelings, Liam (40 years) perhaps found it easier to focus on how he would take his own life. Looking for something to “do the job”, he views it as a simple process, possibly finding a sense of comfort or resolution. He expresses a disconnect between his thinking and emotions—unable to reconcile his chaotic emotions, he switches to a methodical mindset to navigate his way out of his situation.

#### 3.2.2. Unplanned

The attempt being unplanned was evident across half of the interviews (n = 6).


*“I don’t think it I don’t think it was as planned as… as like say for example today’s interview you know it was like… I would say you get up in the morning…and then I just felt horrible all day… I can just remember feeling ehhh no interested in anything at all apart from this thought of… emmm just getting rid of myself basically”*


Robert (49 years) mentions that his attempt was unplanned up until that day where he could not face his negative emotions and situations any longer. The phrase “getting rid of myself” demonstrates how negative his self-perception was at the time, wishing to dispose of himself like you would a piece of rubbish (garbage).

#### 3.2.3. Lived Experience

Many participants (n = 10) had previous experience of suicidal behaviour, either themselves or through friends/family members. Bereavement by suicide was also present—the lasting impact of losing a family member to suicide was significant to participants. Two participants had experienced both previous suicidal behaviour and bereavement or suicidal behaviour in a loved one.

##### Previous Suicidal Behaviour

One of Liam’s (40 years) suicide attempts was characterised by stressful life events and alcohol use (n = 7):


*“I had moved down for a a job…I was far away from family, friends…and… I was having a a bad time…and ehhh I had had ehh it was a very surreal one because I had had a weird funny dream that I had tried to slash my wrists in the bathroom …and then woke up the next morning and walked in and… there was every sharp knife that I owned in the bathroom…and there’s the various things sticking into the floor…and I was like right that wasn’t a dream then…and again I had been drinking for that one”*


Feelings of dissociation are present here; he was unaware of his actions at first (perhaps due to alcohol) and was surprised when he realised what was happening. Being in a dream-like state perhaps was his way of dissociating from the situation he was in.

##### Bereavement/Previous Suicidal Behaviour of a Loved One

Experience of bereavement was something that permeated through the participant’s lives (n = 5). The death of his brother is something that Stephen (45 years) has struggled to come to terms with:


*“yeah… emm yeah because I think we were quite similar because I was was really close to be brother and always looked up to him… I always thought he was he was brilliant… really funny and laughing… so yeah there was that comparison thing well if he’s away then why should I be here, you know what I mean?”*


He holds his brother in such high esteem that he almost feels that because his brother has passed away, he no longer deserves to be alive. He lists all of his brother’s positive qualities. Perhaps he feels like he does not measure up to his brother in this way. 

### 3.3. Dealing with Suicidal Thoughts and Negative Emotions

#### 3.3.1. Avoidance

Avoidance was a strong theme throughout the interviews, with 11 men endorsing this sub-theme. This was significant in many ways; it was used as a coping strategy (e.g. through alcohol use) and as a method to conceal their emotional pain from others.


*“if I think about me… the spider comes in and it grows arms and legs and I’ve got big problems in my head [heid] that aren’t really there in life… but my heid [head] makes them up…and that’s my heid [head] talking to me, trying to get me to go the other way and it’s just about talking about it, trying to skelp it out the way so aye that keeps me going”*


James (31 years) describes feeling out of control, with something other than himself taking over his mind and controlling his actions. He is describing a separate entity to himself, taking over his body (the host), clouding his judgements and taking over his actions, for which he has no control. The spider also has connotations of a fearful image, something to be afraid of. This reflects his perceived inability to cope with his low self-esteem, in combination with his mental illness. Perhaps the only way for him to prevent this happening is to avoid these thoughts altogether?

#### 3.3.2. Seeking Help

Eight men spoke of the recognition that they required help, concerning their mental health as well as other difficult situations in their life, but there were also barriers to this. Stephen (45 years) was desperate for help and tried many avenues: 


*“I mean I used to do everything I started to go to church because I was just so desperate to… and then I started so I was thinking about that and then I was I was scared about going to hell and … just it was running away was like a safety net…you know I was really suicidal there was something I don’t know what it was but something that kept me alive”*


Here, Stephen reports a powerful, instinctual will to stay alive, but he was limited in not knowing where to seek help. The mention of “going to hell” demonstrates the gravity of the predicament he was in, that he felt he had sinned or had done something wrong that he deserved to go to hell. This could also be wrapped up in self-stigmatising attitudes of suicide and suicidal behaviours. Going to church and seeking a higher power to rescue him from his fate that he feels he cannot escape from himself. He may feel condemned in a sense, that he is being punished in life but also risks punishment in death—that there is no escape from his pain and suffering. 

#### 3.3.3. Reached His Limit

A feeling of having no way out, regarding various aspects of their life, was apparent in nine interviews. Many felt that they had struggled too long and it was futile and inescapable. For Blair (28 years), fear of being viewed as a failure led to him to seek a way out of this situation: 


*“… emm it was the pressure and the expectation that I probably put on myself rather than everybody else doing it… thinking there was no way out or if I went home I would be a failure so then rather than being a failure I I wanted to just end it… ill just stop everything…just kind of yeah just have it stop because it was just getting too much”*


The internalised pressure to keep the façade of coping and the intense fear of failing (either in his own eyes or those of others) led to him feeling like the only option was to end his life. In his eyes, seeking help or moving back home was not an option, leading to intense feelings of entrapment. 

### 3.4. Aftermath

#### 3.4.1. Changed but Still Vulnerable 

Following the attempt, the realisation that they have survived the attempt can be puzzling for some (n = 8). Mark (45 years) felt that the suicide attempt had altered his sense of self, and his outlook on life had changed: 


*“I feel better in myself… I feel fragile… I don’t feel… perfect… by any manner of means I’m not I’m not fixed…and I know I’m not fixed… I know I’m not right emm fragile from the point of view that… emm I can just go back into myself…and just bury myself back into myself again…and be quite introverted… I know I could quite easily slip back into that…If I don’t work on it and deal with it … and cope with it”*


It is clear that Mark also does not feel whole again following his suicide attempt—he feels different from the person he was before the attempt. Fragility is a clear notion throughout his interview, his life shattered before and after the attempt in different ways, and he is working towards building himself back up again. The isolation may feel protective as he is still feeling too "fragile" to fully face the world. He uses the term "fixed" to represent a state of being he feels he has not regained since his suicide attempt, but he is unclear exactly what it means to be "fixed".

#### 3.4.2. Altered Sense of Self

Many men (n = 8) struggled to come to terms with the fact that they attempted to take their own life, and their capacity to do so was a shock to many. William (38 years) did not feel like the problems he was facing was worthy of feeling suicidal:


*“I find it embarrassing… yeah…like I don’t have the right… to to do these sorts of things, they’re for really ill people, they’re for people who have real problems and I don’t have them so therefore I’m not entitled to do something like that and… god it is difficult…sorry… and so I’m embarrassed that I I thought I had the right to do that when it’s for someone else”*


Internalised stigma is prevalent across William’s quote. He compares himself to others and judges himself for feeling suicidal. To him, the issues he was facing were insignificant compared to other people and, in his mind, he did not qualify as “really ill”.

### 3.5. Protective Factors

To a lesser extent, the men also detailed factors which had a protective impact for them or provided them with some comfort or support during difficult periods in their life. 

#### 3.5.1. Importance of Talking

Many of the men (n = 10) felt so isolated in the run-up to and following their suicide attempt. As they often did not know how to or did not recognise that they could receive help, having someone approach them first would have been a useful step forward. Blair (28 years) felt that speaking to someone would help him to see the bigger picture:


*“if somebody had spoke then probably yeah… emm… because that distracts you from that thought and you start talking about something else… emm… apart from that probably not much emm unless there was honestly somebody there… ehh but yeah just… having a distraction to take you away from it so you don’t think of… just doing it or you know what can be from somebody else to try to reiterate that you’ve got somebody else to live for or something else to live for…definitely”*


Blair was so consumed by his negative thoughts at the time that having someone there for comfort and to help him recognise the positive aspects of his life would have been valuable. He emphasises the need for a distraction in order to provide a release or escape from his suicidal thoughts at the time. 

#### 3.5.2. Importance of Relationships 

Social connections and relationships with others were an important protective factor for many men (n = 9), particularly feeling valued:


*“well I’m quite glad… that I’m still here from that perspective… especially because I can see that… it’s no [not] just… which is you know it’s no [not] about I can see your worth because…you need me… but at least somebody needs me…I can see that folk actually need me about”*


Mark (45 years) is now able to recognise that he does have a positive impact on other people’s lives and is relied on. He can see more clearly that "folk actually need me about", that he is worthy of life and meaningful relationships.

## 4. Discussion

The factors contributing to the decision to take their own life differed among the men interviewed. However, certain sub-themes and risk factors prevailed across the interviews. There was a notable shift in their pattern of thinking to single mindedness, that once they had decided to take their own life, nothing could stop them. By diverting their mind to the mechanics of attempting to take their own life, it perhaps provided some solace in the face of sometimes chaotic emotions or situations. There was a sudden sense of clarity following a period of cognitive and emotional chaos. The need for help and support was recognised by the interviewees, although some did not know where to access this and did not wish to be viewed as vulnerable or a failure. Further, some participants felt that they had struggled for so long and felt that they could no longer continue to live in their current circumstances. 

Prolonged periods of poor mental health or difficult life circumstances prior to the suicide attempt were evident, in line with previous research [2,22]. This is consistent with the Integrated Motivational Volitional (IMV) model [23], as many men expressed that they had reached their limit, contributing to a sense of entrapment, where they felt suicide was the only option. Further, the discussion of methods may reflect the masculine notion of having an outward display of strength and the desire to avoid being viewed as weak due to a “failed” suicide attempt [4]. This also highlights the importance of attempting to identify potentially vulnerable groups of men before the point at which they’ve expressed suicidal ideation or plans. 

Lived experience of suicide and self-injurious behaviours were prevalent, with some men also being affected by the suicidal behaviour of their loved ones. Substance use was common, which could be associated with access to means or how men cope with suicidal thoughts [22,24,25]. The coping strategies detailed were consistent with Rasmussen, Hjelmeland [10], and Cleary [26], as many men felt that they had to avoid these thoughts altogether perhaps due to a fear of what they could be capable of (attempting to take their own life). This may also be linked to the notion of self-reliance, as many recognised that they did need help but were reluctant or fearful to admit this or did not want to be viewed as a failure [10]. The men also expressed a feeling of pressure to live up to what they felt was a successful male, and a failure to do so resulted in intense feelings of shame [27,28,29]. The social and cultural contexts are also of central importance [18]. For example, many men in this study engaged in avoidant behaviours and appeared to suppress their emotions, perhaps to avoid being viewed as weak and to adhere to perceived cultural norms which discourage disclosure of emotional vulnerability. Examining suicide from a life-course perspective aids understanding of male suicide from different standpoints, such as young people in crisis, mid-life gendered patterns of work and family as well as from the perspective of older adults [17].

The prevailing impact of the suicide attempt was significant—a notion of fragility emanated from the transcripts, with some of the men no longer feeling whole again after their attempt and feeling afraid of going back to that ‘dark’ place once more. This concept of fragility has been noted in previous research [30], perhaps most starkly evident in the fact that men have a lower life expectancy than women in the UK. Further, fragility in males appears to be mediated by social factors [30] such as reduced life chances due to unemployment and addictions, which has been detailed by the men in this study. In terms of protective factors, many men were so isolated in the run-up to and following their suicide attempt that having someone approach them in the first instance would be a useful step forward. In particular. It seems that they were often unsure how to reach out for help or even recognise that they were worthy of help. Social connections and relationships with others were important protective factors for many men, particularly feeling valued [9,13]. It was encouraging that men were open to talking about their problems and emotions in this study. However, there are often structural, emotional or political inequalities and conditions which can shape men’s distress which require more comprehensive investment and societal change beyond talking [31].

### 4.1. Clinical Implications

The findings of this study have some significant clinical implications. Firstly, the findings indicate the difficulties that men experience following their suicide attempt, describing themselves as “fragile” or “in shock”, demonstrating the significant impact that this has on the lives and the need for support during this vulnerable period. By recognising these experiences, it may be possible to better identify those at risk of attempting suicide, particularly because men may not readily discuss emotional problems. The majority of the men acknowledged that they needed help but were either unable to reach out for help or did not know where to seek help. This highlights that men may be ready to seek help and would benefit from friends/family or support services approaching them in the first instance. Further, previous research [5,32] has identified that men may display signs of mental illness differently to women. Many men engaged in avoidant behaviour and recognising that this is a common strategy or manifestation of mental illness in men is a useful step forward in the identification and treatment of men at risk of dying by suicide. 

In the months and years following the suicide attempt, many men struggle to come to terms with the fact that they had attempted to take their own life and felt different to (lesser than) ‘the man’ they were before the attempt. This fragility also conferred risk for future suicide attempts. It also altered their self-image, challenging their view of themselves as well as the notion of the type of person who attempts to take their own life, which is consistent with previous research [33]. Finding ways to bolster their self-esteem and self-image should be an integral aspect of their recovery.

### 4.2. Strengths and Limitations

It is important to take into consideration the limitations of this study when interpreting the findings. This sample includes Scottish men, who are predominantly white and have survived a suicide attempt, hence the findings may not be generalisable to other genders, ethnicities or those who have died by suicide. In order to view the recruitment adverts, potential participants would need access to the internet, which may exclude older or more disadvantaged men. The sample is fairly broad in terms of age group, which allows for a range of perspectives to be included in this study. The idiographic process of IPA allowed for the interviews to be guided by participants, following the topics that were significant to each individual. Each participant’s account may be subject to memory biases, for example recalling negative events more readily than positive events. In line with Emslie, Ridge [34], it was possible to identify an adequate sample of men who were able to talk about their life experiences, mental illness, and suicide, which demonstrates that men are willing to talk about their thoughts and feelings. Hence, the depiction that men who experience depression are silent, is not wholly accurate. Like all such studies, our findings are limited because we cannot speak to the very people who we would like to, namely men who have died by suicide. Nonetheless, our interviews provide important insights into the minds of men who have been suicidal in the past.

## 5. Conclusions

This study explored the suicidal process in men, from suicide attempt to recovery. The findings provide insights into how men cope with suicidal thoughts or negative emotions, often avoiding seeking help and suppressing their emotions. The men’s lives were significantly affected by the attempt, with some stating that they had changed as a person. Importantly, the findings indicate that men do recognise that they need help and can be receptive to help but can feel they need to be approached in the first instance. This offers an encouraging potential opportunity for support networks and clinical services caring for vulnerable men. 

## Figures and Tables

**Table 1 ijerph-18-05209-t001:** Major themes and related sub-themes.

Major Themes	Sub-themes		
**Characteristics of Attempt/** **Volitional Factors**	Change in Thinking	Unplanned	Lived Experience
**Dealing with Suicidal Thoughts/Negative Emotions**	Avoidance	Seeking Help	Reached His Limit
**Aftermath**	Changed but Still Vulnerable	Altered Sense of Self	
**Protective Factors**	Importance of Talking	Importance of Relationships	

## Data Availability

The data presented in this study are available on request from the corresponding author. The data are not publicly available due to the data being held on University Servers in the UK and thus will be subject to appropriate organisational and technical safeguards.

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
