# Peer review of "The Male Experience of Suicide Attempts and Recovery: An Interpretative Phenomenological Analysis"

_ijerph, 2021, doi:10.3390/ijerph18105209_

Round 1

Reviewer 1 Report

The present paper investigates the male subjective experience of suicide attempts and recovery using Interpretative Phenomenological Analysis (IPA).  The methodology is very clearly described, and also a reader that doesn't know this analysis (like me) can clearly get through the paper and understand the results.  The paper is very well written and readable. The topic is extremely important given that men’s deaths by suicide outnumber women worldwide and across every age range. I would just suggest to the authors some more thoughts about distal risk factors for suicide in men, as they talk about "fragility". A very interesting editorial shows that boys' fragility starts very early in life, compared to girls https://www.bmj.com/content/321/7276/1609 .  I believe that findings from the present research have also cultural implications, do we also need a shift on the cultural and developmental level?  Do we need to start considering more male fragility, from the beginning of life? I think the subjective experience of suicide might suggest a new idea of being male. I thank the author in advance for their answer on this aspect, which might be included in the discussion.

Author Response

Comment 1:

The present paper investigates the male subjective experience of suicide attempts and recovery using Interpretative Phenomenological Analysis (IPA).  The methodology is very clearly described, and also a reader that doesn't know this analysis (like me) can clearly get through the paper and understand the results.  The paper is very well written and readable. The topic is extremely important given that men’s deaths by suicide outnumber women worldwide and across every age range.

Author reply 1:

Thank you for reviewing our article and recommending it for publication.

Comment 2:

I would just suggest to the authors some more thoughts about distal risk factors for suicide in men, as they talk about "fragility". A very interesting editorial shows that boys' fragility starts very early in life, compared to girls https://www.bmj.com/content/321/7276/1609 . 

Author reply 2:

Thank you for highlighting this interesting paper, this has now been included in the discussion (line numbers 358-362):

“This concept of fragility has been noted in previous research [30], perhaps most starkly evident in the fact that men have a lower life expectancy than women in the UK. Also, fragility in males appears to be mediated by social factors [30], such as reduced life chances due to unemployment and addictions, which has been detailed by the men in this study.”

Comment 3:

I believe that findings from the present research have also cultural implications, do we also need a shift on the cultural and developmental level?  Do we need to start considering more male fragility, from the beginning of life? I think the subjective experience of suicide might suggest a new idea of being male. I thank the author in advance for their answer on this aspect, which might be included in the discussion.

Author reply 3:

Thank you for your comment. The discussion has been amended (line number 348-355):

“The social and cultural contexts are also of central importance [18]. For example, many men in this study engaged in avoidant behaviours and appeared to suppress their emotions, perhaps to avoid being viewed as weak and to adhere to perceived cultural norms which discourage disclosure of emotional vulnerability. Examining suicide from a life-course perspective aids understanding of male suicide from different standpoints, such as young people in crisis, mid-life gendered patterns of work and family as well as from the perspective of older adults [17].”

Reviewer 2 Report

The authors write about male relatively young suicide attempters recruited through advertisements in social media. An interpretative phenomenological analysis has been used with semi-structured taped interviews. Several researchers have listened to the tapes but the main author performed all the interviews.

The characteristics of the sample are presented in a table and to some extent in the text. I miss a couple of relevant parameters, the most important ones being religion and suicide method. I do not know whether the authors avoid such information for integrity reasons.

The applicability of the findings of course suffers from a major problem : These subjects made attempts that were not « successful ». While it is true that there is a grey zone between successful and non-successful attempts there is still a difference. In general successful attempts are more determined. This needs to be more discussed.

Perhaps the most important observation is that these subjects feel that they became more vulnerable as persons after the suicide attempt. This is an important point which should be emphasised more.

The big table needs some polish

Author Response

Comment 1:

The authors write about male relatively young suicide attempters recruited through advertisements in social media. An interpretative phenomenological analysis has been used with semi-structured taped interviews. Several researchers have listened to the tapes but the main author performed all the interviews.

The characteristics of the sample are presented in a table and to some extent in the text. I miss a couple of relevant parameters, the most important ones being religion and suicide method. I do not know whether the authors avoid such information for integrity reasons.

Author reply 1:

Thank you for your comments. We did not include religion or suicide method in the pre-interview questionnaire, but this is something we will consider for future work.

Comment 2:

The applicability of the findings of course suffers from a major problem : These subjects made attempts that were not « successful ». While it is true that there is a grey zone between successful and non-successful attempts there is still a difference. In general successful attempts are more determined. This needs to be more discussed.

Author reply 2:

The participants in this study were screened and the suicide attempt was discussed with the interviewer (to the extent that they felt comfortable with). Many participants required hospital treatment following their suicide attempt which would classify this as a “serious” attempt. The participants were also screened to determine whether they had attempted to take their own life or whether they had suicidal ideation. Participants were eligible for this study if they had attempted suicide.  However, we have added the following to the discussion on p. 10 (line numbers 408-411):

“Like all such studies, our findings are limited because we cannot speak to the very people who we would like to, namely men who have died by suicide. Nonetheless, our interviews provide important insights into the minds of men who have been suicidal in the past.”

Comment 3:

Perhaps the most important observation is that these subjects feel that they became more vulnerable as persons after the suicide attempt. This is an important point which should be emphasised more.

Author reply 3:

Thank you for your comment and we agree that this is an interesting finding. This has been expanded more in the discussion (line numbers 358-362).

“This concept of fragility has been noted in previous research [30], perhaps most starkly evident in the fact that men have a lower life expectancy than women in the UK. Also, fragility in males appears to be mediated by social factors [30], such as reduced life chances due to unemployment and addictions, which has been detailed by the men in this study.”

Comment 4:

The big table needs some polish

Author reply 4:

Thank you for your comment. The table in Appendix A has been amended to make it clearer and easier to read.